# Private and Robust Federated Learning using Private Information Retrieval and Norm Bounding

**Hamid Mozaffari***
University of Massachusetts Amherst
`hamid@cs.umass.edu`

**Virendra J. Marathe**
Oracle Labs
`virendra.marathe@oracle.com`

**Dave Dice**
Oracle Labs
`dave.dice@oracle.com`

## Abstract

Federated Learning (FL) is a distributed learning paradigm that enables mutually untrusting clients to collaboratively train a common machine learning model. Client data privacy is paramount in FL. At the same time, the model must be protected from poisoning attacks from adversarial clients. Existing solutions address these two problems in isolation. We present FedPerm, a new FL algorithm that addresses both these problems by combining norm bounding for model robustness with a novel intra-model parameter shuffling technique that amplifies data privacy by means of Private Information Retrieval (PIR) based techniques that permit cryptographic aggregation of clients' model updates. The combination of these techniques helps the federation server constrain parameter updates from clients so as to curtail effects of model poisoning attacks by adversarial clients. We further present FedPerm's unique hyperparameters that can be used effectively to trade off computation overheads with model utility. Our empirical evaluation on the MNIST dataset demonstrates FedPerm's effectiveness over existing Differential Privacy (DP) enforcement solutions in FL.

## 1 Introduction

Federated Learning (FL) is a distributed learning paradigm where mutually untrusting *clients* collaborate to train a shared model, called the *global model*, without explicitly sharing their local training data. FL training involves a *server* that aggregates, using an *aggregation rule* (AGR), model updates that the clients compute using their local private data. The aggregated *global model* is subsequently broadcasted by the server to a subset of the clients. This process repeats for several rounds until convergence or a threshold number of rounds. Though highly promising, FL faces multiple challenges [25] to its practical deployment. Two of these challenges are (i) data privacy for clients' training data, and (ii) robustness of the global model in the presence of malicious clients.

The data privacy challenge emerges from the fact that raw model updates of federation clients are susceptible to privacy attacks by an adversarial server as demonstrated by several recent works [28, 29, 37, 49, 54]. Two classes of approaches can address this problem in significantly different ways: First, *Local Differential Privacy* [16, 26, 45, 48] in FL (LDP-FL) enforces a strict theoretical privacy guarantee to model updates of clients. The guarantee is enforced by applying carefully calibrated noise to the clients' local model updates using a local randomizer $\mathcal{R}$. In addition to the privacy guarantee, LDP-FL can defend against poisoning attacks by malicious clients, thus providing robustness to the

---

*The work was done while interning at the Oracle Labs.

Workshop on Federated Learning: Recent Advances and New Challenges, in Conjunction with NeurIPS 2022 (FL-NeurIPS'22). This workshop does not have official proceedings and this paper is non-archival.

global model [38, 36, 43]. However, the model update perturbation needed for the LDP guarantee significantly degrades model utility.

The other approach to enforce client data privacy is *secure aggregation (sAGR)*, where model update aggregation is done using cryptographic techniques such as homomorphic encryption or secure multi-party computation [11, 53, 8, 21]. sAGR protects privacy of clients' data from an adversarial server because the server sees just the encrypted version of clients' model updates. Moreover, this privacy is enforced without compromising global model utility. However, the encrypted model updates themselves provide the perfect cover for a malicious client to poison the global model [21, 38] – the server cannot tell the difference between a honest model update and a poisoned one since both are encrypted.

In this paper we answer the dual question: *Can we design an efficient federated learning algorithm that achieves local privacy for participating clients at a low utility cost, while ensuring robustness of the global model from malicious clients?* To that end, we present *FedPerm*, a new FL protocol that combines LDP [16, 26, 48], model parameter shuffling [19], and *computational Private Information Retrieval (cPIR)* [14, 13, 1, 2] in a novel way to achieve our dual goals.

The starting point of FedPerm's design is *privacy amplification by shuffling* [19], which enables stronger (i.e., amplified) privacy with little model perturbation (using randomizer $\mathcal{R}$) at each client. Crucially, our shuffling technique fundamentally differs from prior works in that we apply *intra-model* parameter shuffling rather that the *inter-model* parameter shuffling done previously [19, 30, 24].

Next, each FedPerm client privately chooses its *shuffling pattern* uniformly at random for each FL round. To aggregate the shuffled (and perturbed) model parameters, FedPerm client utilizes cPIR to generate a set of PIR queries for its shuffling pattern that allows the server to retrieve each parameter *privately* during aggregation. All the server observes is the shuffled parameters of the model update for each participating client, and a series of PIR queries (i.e., the encrypted version of the shuffling patterns). The server can aggregate the PIR queries and their corresponding shuffled parameters for multiple clients to get the encrypted aggregated model. The aggregated model is decrypted independently at each client.

The combination of LDP at each client and intra-model parameter shuffling achieves enough privacy amplification to let FedPerm preserve high model utility. At the same time, availability of the shuffled parameters at the federation server lets it control a client's model update contribution by enforcing norm-bounding, which is known to be highly effective against model poisoning attacks [38, 36, 43].

Since FedPerm utilizes cPIR which relies on homomorphic encryption (HE) [39, 15], it can be computationally expensive, particularly for large models. We present computation/utility trade off hyper-parameters in FedPerm, that enables us to achieve an interesting trade off between computational efficiency and model utility. In particular, we can adjust the computation burden for a proper utility goal by altering the size and number of shuffling patterns for the FedPerm clients.

We empirically evaluate FedPerm on the MNIST dataset to demonstrate that it is possible to provide LDP-FL guarantees at low model utility cost. We theoretically and numerically demonstrate a trade off between model utility and computational efficiency. Specifically, FedPerm's hyperparameters create *shuffling windows* whose size can be reduced to drastically cut computation overheads, but at the cost of reducing model utility due to lower privacy amplification. We experiment with two representative shuffling window configurations in FedPerm– "light" and "heavy". For a $(4.0, 10^{-5})$-LDP guarantee, the light version of FedPerm, where client encryption, and server aggregation needs 52.2 seconds and 21 minutes respectively, results in a model that delivers 32.85% test accuracy on MNIST. The heavier version of FedPerm, where client encryption and server aggregation needs 32.1 minutes and 16.4 hours respectively, results in 72.38% test accuracy. Non-private FedAvg, CDP-FL and LDP-FL provide 91.02%, 53.50%, and 13.74% test accuracies for the same $(\varepsilon, \delta)$-DP guarantee respectively.

## 2 Preliminaries

In FL [32, 25, 27], $N$ clients collaborate to train a global model without directly sharing their data. In round $t$, the federation server samples $n$ out of $N$ total clients and sends them the most recent global model $\theta^t$. Each client re-trains $\theta^t$ on its private data using stochastic gradient descent (SGD), and sends back the model parameter updates ($x_i$ for $i^{th}$ client) to the server. The server then aggregates (e.g., averages) the collected parameter updates and updates the global model for the next round ($\theta^t \leftarrow \theta^{t-1} + \frac{1}{n} \sum_{i=1}^{n} x_i$).

**Central Differential Privacy in FL (CDP-FL).** In CDP-FL [12, 22], illustrated in Figure 1(a), a *trusted* server first collects all the clients' raw model updates ($x_i \in \mathbb{R}^d$), aggregates them into the global model, and then perturbs the model with carefully calibrated noise to enforce differential privacy (DP) guarantees. The server provides participant-level DP by the perturbation. We defer the definition and algorithm of CDP-FL to Appendix E.1.

**Local Differential Privacy in FL (LDP-FL).** CDP-FL relies on availability of a trusted server for collecting raw model updates. On the other hand, LDP-FL [31, 47] does not rely on this assumption and each client perturbs its output locally using a randomizer $\mathcal{R}$ (Figure 1(b)). If each client perturbs its model updates locally by $\mathcal{R}$ which satisfies $(\varepsilon_\ell, \delta_\ell)$-LDP, then observing collected updates $\{\mathcal{R}(x_1), \ldots, \mathcal{R}(x_n)\}$ also implies $(\varepsilon_\ell, \delta_\ell)$-DP [17].

**Definition 2.1 (Local Differential Privacy (LDP))** *A randomized mechanism $\mathcal{R} : \mathcal{X} \to \mathcal{Y}$ is said to be $(\varepsilon_\ell, \delta_\ell)$-locally differentially private if for any two inputs $x, x' \in \mathcal{X}$ and any output $y \in \mathcal{Y}$, we have $Pr[\mathcal{R}(x) = y] \leq e^{\varepsilon_\ell} \Pr[\mathcal{R}(x') = y] + \delta_\ell$.*

In LDP-FL, each client perturbs its local update ($x_i$) with $\epsilon_\ell$-LDP. Unfortunately, LDP hurts utility, especially for high dimensional vectors. Its mean estimation error is bounded by $O(\frac{\sqrt{d \log d}}{\varepsilon_\ell \sqrt{n}})$ meaning that for better utility we should increase the privacy budget or use larger number of users in each round [9].

## 2.1 Privacy Amplification by Shuffling Clients' updates

Recent works [24, 30] utilize the privacy amplification effect by shuffling model parameters across client model updates from participating clients to improve the LDP-FL utility (illustrated in Figure 1(c)). FL frameworks based on shuffling clients' updates consists of three building processes: $\mathcal{M} = \mathcal{A} \circ \mathcal{S} \circ \mathcal{R}$. Specifically, they introduce a shuffler $\mathcal{S}$, which sits between the FL clients and the FL server, and randomly shuffles parameters across clients' locally perturbed updates (by randomizer $\mathcal{R}$) before sending them to the server for aggregation ($\mathcal{A}$). More specifically, given parameter index $i$, $\mathcal{S}$ randomly shuffles the $i^{th}$ parameters of model

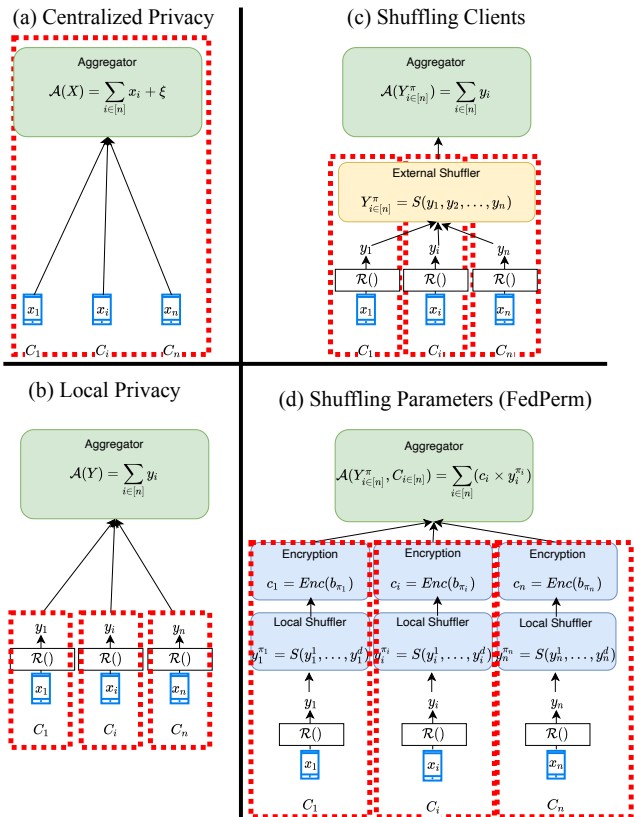

Figure 1: Different models of differential privacy in Federated Learning. Red dots are showing the trust boundaries.

updates received from the $n$ participant clients. The shuffler thus detaches the model updates from their origin client (i.e. anonymizes them). Previous works [4, 5, 23] focused on shuffling one-dimensional data $x \in X$. Corollary 2.1 shows the privacy amplification effect by shuffling.

**Corollary 2.1** *[6] In shuffle model, if $\mathcal{R}$ is $\varepsilon_\ell$-LDP, where $\varepsilon_\ell \leq \log(n/\log(1/\delta_c))/2$. $\mathcal{M}$ satisfies $(\varepsilon_c, \delta_c)$-DP with $\varepsilon_c = O((1 \wedge \varepsilon_\ell)e^{\varepsilon_\ell}\sqrt{\log(1/\delta_c)/n})$ where '$\wedge$' shows minimum function.*

From above corollary, the privacy amplification has a direct relationship with $\sqrt{n}$ where $n$ is the number of selected clients for aggregation, i.e., increasing the number of clients will increase the privacy amplification. Note that in FedPerm, the clients are responsible for shuffling, and instead of shuffling the $n$ clients' updates (inter-model shuffling), each client locally shuffles its $d$ parameters (intra-model shuffling). In real-world settings there is a limit on the value of $n$, so the amount of amplification we can achieve is also limited. However, in FedPerm we can see much more amplification because we are shuffling the parameters and $n \ll d$.

We present an overview of central differential privacy in FL (CDP-FL), Laplace mechanism, privacy composition theorems, robustness to poisoning attacks, private information retrieval (PIR), and homomorphic encryption systems in Appendix E.

## 3 FedPerm: Private and Robust Federated Learning by parameter Permutation

We assume a dual threat model setting where (i) the federation server acts as an *honest but curious* aggregator, and (ii) the federation clients can maliciously attempt to poison the trained model using manipulated local parameter updates. We provide further details about the threat models in Appendix D.

### 3.1 FedPerm: Design

FedPerm utilizes computational Private Information Retrieval (cPIR) [14, 42] for secure aggregation at the federation server. In particular, FedPerm uses the cPIR algorithm by Chang [13] that leverages the algorithm by Paillier [39]. Algorithm 1 depicts FedPerm. Figure 1(d) depicts the FedPerm framework that consists of three components, $\mathcal{F} = \mathcal{A} \circ \mathcal{S} \circ \mathcal{R}^d$, denoting the client-side parameter randomizer ($\mathcal{R}^d$), the client-side shuffler ($\mathcal{S}$), and the server-side aggregator ($\mathcal{A}$).

**Key Distribution** Paillier is a Partial Homomorphic Encryption (PHE) algorithm that relies on a public key encryption scheme (details of Paillier HE in Appendix E.6). Since Paillier is employed to protect client updates from a curious federation server, FedPerm requires an independent key server that generates a pair of public and secret homomorphic keys $(Pk, Sk)$. This key pair is distributed to all federation clients, and just the public key $Pk$ is sent to the federation server (for aggregation). The key server itself can be implemented as an independent third party server, or a leader among the federation clients may be chosen to play that role [53].

**Client Local Training:** In the $t^{th}$ round, the server randomly samples $n$ clients among total $N$ clients. Each sampled client locally retrains a copy of the global model it receives from the server ($\theta_g^t$), optimizing the model using its local data and local learning rate $\eta$ (Algorithm 1, line 5).

**Randomizing Update Parameters:** After computing local updates $\theta_u^t$, client $u$ clips the update using threshold $C$ and normalizes the parameters to the range $[0, 1]$ (Algorithm 1, lines 6-7). Now the client applies the randomizer (i.e., $\mathcal{R}^d$) on its local parameters to make them ($\varepsilon_d$)-differentially private (Algorithm 1, line 8). We use the Laplacian Mechanism as a local randomizer with privacy budget $\varepsilon_d$.

**Shuffling:** After clipping and perturbing the local update, each client shuffles the parameters $y_u^t$ using the random shuffling pattern $\pi_u$ (Algorithm 1, lines 9-10). Shuffling amplifies the privacy budget $\varepsilon_d$, which we discuss in Section 3.2.

**Generating PIR queries:** Now the client encodes the shuffle indices $\pi_u$ using a PIR protocol. This process comprises two steps: first creating a binary mask of the shuffled index, and then encrypting it using the public key of HE that the client received in first step (Algorithm 1 line 11-12). Generally, a PIR client needs access to the $j^{th}$ record privately from an untrusted PIR server that holds a dataset $\theta$ with $d$ records; i.e. the PIR server cannot know that the client re-

**Algorithm 1** FedPerm where green and blue colors show execution by server and client respectively.

**Input**: number of FL rounds $T$, number of local epochs $E$, number of selected users in each round $n$, learning rate $\eta$, local privacy budget $\varepsilon_d$, number of model parameters $d$, parameter update clipping threshold $C$
**Output**: $\theta_g^T$

1:   $\theta_g^0 \leftarrow$ Initialize weights
2: **for** each iteration $t \in [T]$ **do**
3:     $U \leftarrow$ set of $n$ randomly selected clients out of $N$ total clients
4:     **for** $u$ in $U$ **do**
5:       $\theta_u^t \leftarrow$ LOCALUPDATE$(\theta_g^t, \eta, E)$
6:       $\bar{\theta}_u^t \leftarrow$ CLIP$(\theta_u^t, -C, C)$
7:       $\tilde{\theta}_u^t \leftarrow (\bar{\theta}_u^t + C)/(2C)$
8:       $y_u^t \leftarrow$ RANDOMIZE$(\tilde{\theta}_u^t, \varepsilon_d)$
9:       $\pi_u \leftarrow$ Shuffling pattern RANDOMPERMUTATIONS $\in [1, d]$
10:      $\tilde{y}_u^t \leftarrow$ SHUFFLE$(y_u^t, \pi_u)$
11:      $b_u^t \leftarrow$ BINARYMASK$(\pi_u)$
12:      $c_u^t \leftarrow$ ENC$_{pk}(b_u^t)$
13:      Client $u$ sends $(\tilde{y}_u^t, c_u^t)$ to the server
14:     **end for**
15:     norm bounding: $\tilde{y}_u^t \leftarrow \tilde{y}_u^t \cdot \min(1, \frac{M}{||\tilde{y}_u^t||_2})$ for $u \in U$
16:     $\bar{z} \leftarrow \frac{1}{n} \sum_{u \in U} (c_u^t \times \tilde{y}_u^t)$
17:     $z \leftarrow$ DEC$_{sk}(\bar{z})$
18:     normalize $z \leftarrow C \cdot (2z - 1)$
19:     update model $\theta_g^{t+1} \leftarrow \theta_g^t + z$
20: **end for**
21: **return** $\theta_g^T$

quested the $j^{th}$ record. To do so, the PIR client
creates a unit vector (binary mask) $\vec{b}_j$ of size $d$ where all the bits are set to zero except the $j^{th}$ position
being set to one (i.e., $\vec{b}_j = [0 \quad 0 \quad \ldots \quad 1 \quad \ldots \quad 0 \quad 0]$).

If the PIR client does not care about privacy, it would send $\vec{b}_j$ to the PIR server, and the server would
generate the client's response by multiplying the binary mask into the database matrix $\theta$ ($\theta_j = \vec{b}_j \times \theta$).
A PIR technique allows the client to obtain this response without revealing $\vec{b}_j$ to the PIR server. For
example in [13], the PIR client uses HE to encrypt $\vec{b}_j$ element by element before sending it to the
PIR server. During the data recovery phase, the client extracts its target record by decrypting the
component of $\text{ENC}(\vec{b}_j) \times \theta$. Equation 1 shows retrieving the $j^{th}$ record by this PIR query. Note that
an HE system has a property that $m_1 \times m_2 \leftarrow \text{DEC}(\text{ENC}[m_1] \times m_2)$.

$$\text{DEC}(\text{ENC}(\vec{b}_j) \times \theta) = \text{DEC}(\text{ENC}[0] \cdot \theta_1 + \cdots + \text{ENC}[1] \cdot \theta_j + \cdots + \text{ENC}[0] \cdot \theta_d) = \text{DEC}(\text{ENC}[\theta_j]) = \theta_j \tag{1}$$

A FedPerm client creates $d$ PIR queries to retrieve each parameter privately. (In Section 3.2, we
discuss additional parameters to reduce the number of PIR queries.) In this case, the shuffled
parameters ($\tilde{y}_u^t$) are the dataset located at the PIR server and each shuffled index in $\pi_u$ is the secret
record row number (i.e. $j^{th}$ in above) that the PIR client is interested in. Client $u$ first creates $b_u^t$
which is a collection of $d$ binary masks of shuffled indices in $\pi_u$, similar to PIR query $\vec{b}_j$. Then the
client encrypts the binary masks and sends the shuffled parameters and the PIR query (encrypted
binary masks) to the server for aggregation.

**Server: norm bounding** After collecting all the local updates $(\tilde{y}_u^t, c_u^t)$ for selected clients in round
$t$, the FedPerm server first applies $\ell_2$-norm bounding to the threshold $M$ on the shuffled parameters
$\tilde{y}_u^t$ (Algorithm 1, line 15). Note that unlike other robust AGRs, *norm bounding* is the only robust
AGR scheme that does not require the true position of the parameters because it works by calculating
the $\ell_2$ norm of the parameter updates as a whole irrespective of their order (i.e. $\ell_2(\tilde{y}_u^t) = \ell_2(y_u^t)$).
Prior works [38, 36, 43] have shown the effectiveness of norm bounding in defense against poisoning
attacks by malicious clients.

**Server: Aggregation** Then the server aggregates all the updates into global update $\bar{z}$ (Algorithm 1,
line 16). This aggregation is averaging the update parameters for $n$ collected updates by calculating
$\frac{1}{n} \sum_{u \in U} (c_u^t \times \tilde{y}_u^t)$. The expression $c_u^t \times \tilde{y}_u^t$ has the effect of "unshuffling" client $u$'s parameters. At
the same time, the resulting vector is encrypted, thus kept hidden from the server. In Appendix B, we
show the correctness of FedPerm.

**Updating Global Model** The server aggregates local updates $(\tilde{y}_u^t, c_u^t)$ without knowing the
true position of the parameters as they are detached from their positions. Result of aggregation
$\frac{1}{n} \sum_{u \in U} (c_u^t \times \tilde{y}_u^t)$ is vector of encrypted parameters, and they need to be decrypted to be used for
updating the global model (Algorithm 1 lines 17-19). This decryption is done at each client using
Paillier's secret key.

### 3.2 Computation/Communication and Utility Tradeoff in FedPerm

Each FedPerm client perturbs its local update (vector $x_i$ containing $d$ parameters) with randomizer
$\mathcal{R}^d$ which is $\varepsilon_d$-LDP, and then shuffles its parameters. We use the Laplacian mechanism as the
randomizer. Based on the näive composition theorem from Lemma E.1, the client perturbs each
parameter value with $\mathcal{R}$ which satisfies $\varepsilon_{wd}$-LDP where $\varepsilon_{wd} = \frac{\varepsilon_d}{d}$ (Appendix E.2 contains additional
details). Corollary 3.1 shows the privacy amplification from $\varepsilon_d$-LDP to $(\varepsilon_\ell, \delta_\ell)$-DP after the parameter
shuffling. Corollary 3.1 is derived from Corollary 2.1, by substituting the number of participating
clients $n$ by the number of parameters $d$ in the model.

**Corollary 3.1** *If $\mathcal{R}$ is $\varepsilon_{wd}$-LDP, where $\varepsilon_{wd} \leq \log\left(d/\log\left(1/\delta_\ell\right)\right)/2$, FedPerm $\mathcal{F} = \mathcal{A} \circ \mathcal{S}_d \circ \mathcal{R}^d$
satisfies $(\varepsilon_\ell, \delta_\ell)$-DP with $\varepsilon_\ell = O((1 \wedge \varepsilon_{wd})e^{\varepsilon_{wd}}\sqrt{\log\left(1/\delta_\ell\right)/d})$.*

Thus, larger the number of parameters in the model, greater is the privacy amplification. With large
models containing millions or billions of parameters, the privacy amplification can be immense.
However, the model dimensionality also affects the computation (and communication) cost in
FedPerm. Each FedPerm client generates a $d$-dimensional PIR query for every parameter in the model,
resulting in a PIR query matrix containing $d^2$ entries. This results in a quadratic increase in client
encryption time, server aggregation time, and client-server communication bandwidth consumption.

This increase in communication, and more importantly computation, resources is simply infeasible for large models containing billions of parameters. To address this problem, FedPerm introduces following hyperparameters that present an interesting trade off between computation/communication overheads and model utility:

- **FedPerm with Smaller Shuffling Pattern** *($k_1$):* Instead of shuffling all the $d$ parameters, the FedPerm client can partition its parameters into several identically sized windows, and shuffle the parameters in each window with the *same* shuffling pattern. Thus, instead of creating a very large random shuffling pattern $\pi$ with $d$ indices (i.e., $\pi = \text{RANDOMPERMUTATIONS}[1, d]$), each client creates a shuffling pattern with $k_1$ indices (i.e., $\pi = \text{RANDOMPERMUTATIONS}[1, k_1]$), and shuffles ($\mathcal{S}_{k_1}$) each window with these random indices. The window size $k_1$ is a new FedPerm hyperparameter that can be used to control the computation/communication and model utility trade off. Once we set the size of shuffling pattern to $k_1$, each client needs to perform $d \cdot k_1$ encryptions and consumes $O(d \cdot k_1)$ network bandwidth to send its PIR queries to the server.
- **FedPerm with Multiple Shuffling Patterns** *($k_2$):* An additional way to adjust the computation/communication vs. utility trade off is by using multiple shuffling patterns. Each Fed-Perm client chooses $k_2$ shuffling patterns $\{\pi_1, \dots, \pi_{k_2}\}$ uniformly at random where each $\pi_i = \text{RANDOMPERMUTATIONS}[1, k_1]$ for $1 \le i \le k_2$. Then, each FedPerm client partitions the $d$ parameters into $d/k_1$ windows, where it permutes the parameters of window $k$ ($1 \le k \le d/k_1$) with shuffling pattern $\pi_i$ s.t. $i = k \mod k_2$. In this case, each FedPerm client needs $k_2 \cdot k_1^2$ encryptions to generate the PIR queries.

Due to space limitation, we defer the computation/communication and privacy analysis of these hyperparameters to Appendix A.

## 4 Experiments

In this section, we investigate the utility and computation trade offs in FedPerm. We use MNIST dataset and a logistic regression model with $d = 7850$ parameters to evaluate these trade offs. We compare our results with following baselines: **(a)** FedAvg [32] with no privacy, **(b)** CDP-FL [12, 22], **(c)** LDP-FL [31, 47] with Gaussian Mechanism.

Figure 2 shows the test accuracy of the model trained using different FL algorithms running for $T = 50$ rounds. The MNIST dataset is divided across $n = 15$ clients with a Dirichlet distribution. We compared two versions of FedPerm in these experiments: **(a)** FedPerm with $k_1 = 400$ and $k_2 = 1$ which is a "light" version where encryption and decryption time at clients takes around $52.2$ and $2.4$ seconds respectively. It also imposes $21$ minutes computation time at the server. **(b)** FedPerm with $k_1 = 800$ and $k_2 = 10$ which is a "heavy" version where client encryption, decryption, and server aggregation time takes around $32.1$ minutes, $2.4$ seconds and $16.4$ hours respectively.

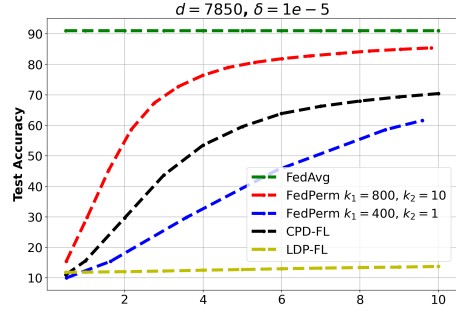

Figure 2: Test accuracy for different FL algorithms for MNIST over 15 clients.

As we mentioned earlier, FedPerm provides a trade-off between privacy amplification and compute resources – larger the values of $k_1$ and $k_2$, greater are the compute resources for training, which in turn provides higher privacy amplification that results in better model utility. The heavy version of FedPerm needs more resources to be as fast as the lighter version, but it can provide much more utility (because the privacy amplification is larger so the amount of noise added is smaller). For instance, after $T = 50$, and total privacy budget $(4.0, 1e^{-5})$, the heavy version provides $72.38\%$ test accuracy while the light version provides $32.85\%$ test accuracy. From these figures we can see if we invest enough computation resources in FedPerm, we can provide higher utility compared to CDP-FL, without trusting the FedPerm server. Non-private FedAvg, CDP-FL and LDP-FL also provides $91.02\%$, $53.50\%$, and $13.74\%$ test accuracies for the same total $(\varepsilon, \delta)$ respectively.

**Miscellaneous Discussions** Due to space limitations, we defer detailed discussion of ablation studies of FedPerm to Appendix. In Appendix C.1, we evaluate the impacts of our hyperparameters $k_1$, $k_2$, $n$, $d$ on the encryption, decryption and server aggregation time. In Appendix A.3, we show the relationship of our hyperparameters on the privacy amplification of FedPerm.

## 5  Conclusion

We presented FedPerm, a new FL algorithm that combines LDP, intra-model parameter shuffling at the federation clients, and a cPIR based technique for parameter aggregation at the federation server to deliver both client data privacy and robustness from model poisoning attacks. Our intra-model parameter shuffling significantly amplifies the LDP guarantee for clients' training data. The cPIR based technique we employ allows cryptographic parameter aggregation at the server. At the same time, the server clips the clients' parameter updates to ensure that model poisoning attacks by adversarial clients are effectively thwarted. We leave the study of extensions to FedPerm – (i) an additional dimension of the hyperparameters ($k_3$) that takes the computation-utility trade offs to hypercube space (see Appendix F), (ii) plugging in other PIR protocols, and (iii) combining an external client shuffler with FedPerm – to future work.

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

## A   Computation/Communication and Utility Tradeoff in FedPerm

In Section 3.2, we show the quadratic increase in client encryption time, server aggregation time for larger number of model parameters $d$. This increase in communication, and more importantly computation, resources is simply infeasible for large models containing billions of parameters. To address this problem FedPerm introduces additional hyperparameters that present an interesting trade off between computation/communication overheads and model utility.

### A.1   FedPerm with Smaller Shuffling Pattern

Instead of shuffling all the $d$ parameters, the FedPerm client can partition its parameters into several identically sized windows, and shuffle the parameters in each window with the *same* shuffling pattern. Thus, instead of creating a very large random shuffling pattern $\pi$ with $d$ indices (i.e., $\pi = \text{RANDOMPERMUTATIONS}[1, d]$), each client creates a shuffling pattern with $k_1$ indices (i.e., $\pi = \text{RANDOMPERMUTATIONS}[1, k_1]$), and shuffles ($\mathcal{S}_{k_1}$) each window with these random indices.

The window size $k_1$ is a new FedPerm hyperparameter that can be used to control the computation/communication and model utility trade off. Once we set the size of shuffling pattern to $k_1$, each client needs to perform $d \cdot k_1$ encryptions and consumes $O(d \cdot k_1)$ network bandwidth to send its PIR queries to the server.

**Superwindow:** A shuffling window size of $k_1$, partitions each FedPerm client $u$'s local update $x_u$ ($d$ parameters) into $w = d/k_1$ windows, each containing $k_1$ parameters. Each FedPerm client, independently from other FedPerm clients, chooses its shuffling pattern $\pi$ uniformly at random with indices $\in [1, k_1]$, and shuffles each window with this pattern. This means that every position $j$ ($1 \leq j \leq k_1$) in each window $k$ ($1 \leq k \leq w$) will have the same permutation index ($\pi_j$). Thus all of the $j^{th}$ positioned parameters ($x_u^{(k,j)}$ for $1 \leq k \leq w$) will contain the value from the $\pi_j^{th}$ slot in window $k$. For a given index $j$ ($1 \leq j \leq k_1$), we define a *superwindow* as the set of all of the parameters $x_u^{(k,j)}$ for $1 \leq k \leq w$. If we structure the parameter vector $x_u$ (with $d$ parameters) as $\mathbb{R}^{k_1 \times w}$ (a matrix with $k_1$ rows and $w$ columns), each row of this matrix is a superwindow.

Figure 3 depicts an example model containing 12 parameters $\theta = [\theta_1, \theta_2, ..., \theta_{12}]$. The original FedPerm algorithm mandates a shuffling pattern $\pi$ with 12 indices $\in [1, 12]$, where the PIR query generates $12 \times 12 = 144$ encryptions. However, a shuffling pattern $\pi$ of three indices $k_1 = 3$ ($\pi = [3, 1, 2]$ in the figure) requires only $3 \times 3 = 9$ encryptions. This shuffling pattern creates 4 windows of size 3 (red boxes in the 2-D matrix in the figure), and each row in the 2-D matrix, represented more succinctly by $[\Theta_A, \Theta_B, \Theta_C]$, itself constitutes a superwindow. The shuffling pattern $\pi = [3, 1, 2]$ applied to $\theta = [\Theta_A, \Theta_B, \Theta_C]$ swaps entire superwindows to give $\mathcal{S}_{k_1}(\theta) = [\Theta_C, \Theta_A, \Theta_B]$.

$$\theta = \begin{bmatrix} \Theta_A \\ \Theta_B \\ \Theta_C \end{bmatrix} = \begin{bmatrix} \theta_1 \\ \theta_2 \\ \theta_3 \end{bmatrix} \begin{bmatrix} \theta_4 \\ \theta_5 \\ \theta_6 \end{bmatrix} \begin{bmatrix} \theta_7 \\ \theta_8 \\ \theta_9 \end{bmatrix} \begin{bmatrix} \theta_{10} \\ \theta_{11} \\ \theta_{12} \end{bmatrix} \text{ where } \pi = \begin{bmatrix} 3 \\ 1 \\ 2 \end{bmatrix}$$

Figure 3: FedPerm example with $k_1 = 3$ and $d = 12$. The red boxes are showing the windows that the parameters inside them are going to be shuffled with the same shuffling pattern $\pi$.

Shuffling of superwindows, instead of individual parameters, leads to a significant reduction in the computation (and communication) overheads for FedPerm clients. However, this comes at the cost of smaller privacy amplification. Corollary A.1 shows the privacy amplification of FedPerm from $\varepsilon_d$-LDP to $(\varepsilon_\ell, \delta_\ell)$-DP after superwindow shuffling (with window size $k_1$). After applying the randomizer $\mathcal{R}$ that is $\varepsilon_d$-LDP on the local parameters, each superwindow is $\varepsilon_w$-LDP where $\varepsilon_w = w \cdot \varepsilon_{wd} = \frac{d}{k_1} \cdot \varepsilon_{wd} = \frac{\varepsilon_d}{k_1}$. Since we are shuffling the superwindows, we can derive Corollary A.1 for FedPerm by setting the shuffling pattern size to $k_1$ from Corollary 2.1.

**Corollary A.1** *For FedPerm $\mathcal{F} = \mathcal{A} \circ \mathcal{S}_{k_1} \circ \mathcal{R}^w$ with window size $k_1$, where $\mathcal{R}^w$ is $\varepsilon_w$-LDP and $\varepsilon_w \leq \log{(k_1 / \log{(1/\delta_\ell)})}/2$, the amplified privacy is $\varepsilon_\ell = O((1 \wedge \varepsilon_w)e^{\varepsilon_w}\sqrt{\log{(1/\delta_\ell)}/k_1}$.*

### A.2 FedPerm with Multiple Shuffling Patterns

An additional way to adjust the computation/communication vs. utility trade off is by using multiple shuffling patterns. Each FedPerm client chooses $k_2$ shuffling patterns $\{\pi_1, \ldots, \pi_{k_2}\}$ uniformly at random where each $\pi_i = \text{RANDOMPERMUTATIONS}[1, k_1]$ for $1 \leq i \leq k_2$. Then, each FedPerm client partitions the $d$ parameters into $d/k_1$ windows, where it permutes the parameters of window $k$ ($1 \leq k \leq d/k_1$) with shuffling pattern $\pi_i$ s.t. $i = k \mod k_2$. In this case, each FedPerm client needs $k_2 \cdot k_1^2$ encryptions to generate the PIR queries.

Figure 4 shows FedPerm for $d = 12$, $k_1 = 3$ and $k_2 = 2$, i.e., there are two shuffling patterns $\pi_1$ (shown with red box) and $\pi_2$ (shown with blue box) and each one has 3 shuffling indices. In this example, the client partitions the 12 parameters into 4 windows that it shuffles with $\pi_1$ (1st and 3rd windows) and $\pi_2$ (2nd and 4th windows). This example is equivalent to an FL scenario with *two* external inter-model shufflers (with shuffling patterns $\pi_1, \pi_2$) and three FL clients $(A, B, C)$. Each client sends 2 ($w = d/(k_1 k_2)$) parameters to each shuffler for shuffling with other clients. Two different shuffling patterns $\pi_1$ and $\pi_2$ are applied on $[\Theta_{A1}, \Theta_{B1}, \Theta_{C1}]$ and $[\Theta_{A2}, \Theta_{B2}, \Theta_{C2}]$ respectively.

$$\theta = \begin{bmatrix} \Theta_{A1} & \Theta_{A2} \\ \Theta_{B1} & \Theta_{B2} \\ \Theta_{C1} & \Theta_{C2} \end{bmatrix} = \begin{bmatrix} \theta_1 \\ \theta_2 \\ \theta_3 \end{bmatrix} \begin{bmatrix} \theta_4 \\ \theta_5 \\ \theta_6 \end{bmatrix} \begin{bmatrix} \theta_7 \\ \theta_8 \\ \theta_9 \end{bmatrix} \begin{bmatrix} \theta_{10} \\ \theta_{11} \\ \theta_{12} \end{bmatrix} = \begin{bmatrix} \theta_1 & \theta_7 \\ \theta_2 & \theta_8 \\ \theta_3 & \theta_9 \end{bmatrix} \begin{bmatrix} \theta_4 & \theta_{10} \\ \theta_5 & \theta_{11} \\ \theta_6 & \theta_{12} \end{bmatrix}$$

$$\text{where } \pi_1 = \begin{bmatrix} 3 \\ 1 \\ 2 \end{bmatrix} \quad \pi_2 = \begin{bmatrix} 2 \\ 1 \\ 3 \end{bmatrix}$$

Figure 4: FedPerm example with $k_1 = 3$ and $k_2 = 2$. We have two shuffling patterns $\pi_1$ and $\pi_2$ shown with red and blue boxes.

When we have $k_2$ shuffling patterns and each shuffling pattern has $k_1$ indices, the size of each superwindow is $w = d/(k_1 k_2)$. Therefore, each client perturbs each superwindow with a randomizer $\mathcal{R}^w$ that satisfies $\varepsilon_w$-LDP where $\varepsilon_w = w \cdot \varepsilon_{wd} = \frac{d}{k_1 k_2} \cdot \varepsilon_{wd} = \frac{\varepsilon_d}{k_1 k_2}$. Take $\varepsilon_w$ to Corollary 2.1 on the superwindows to find the amplified local privacy and then using strong composition in Lemma E.2 we can easily derive the Theorem A.2 for FedPerm with $\mathcal{S}_{k_1}^{k_2}$.

**Theorem A.2** *For FedPerm $\mathcal{F} = \mathcal{A} \circ \mathcal{S}_{k_1}^{k_2} \circ \mathcal{R}^w$ with window size $k_1$, and $k_2$ shuffling patterns, where $\mathcal{R}^w$ is $\varepsilon_w$-LDP and $\varepsilon_w \leq \log\left(k_1 / \log\left((k_2 + 1)/\delta_\ell\right)\right)/2$, the amplified privacy is $\varepsilon_\ell = O((1 \wedge \varepsilon_w)e^{\varepsilon_w} \log\left(k_2/\delta_\ell\right)\sqrt{k_2/k_1})$.*

### A.3   Privacy Analysis

In Figure 5, we show the relationship of our introduced variables $k_1$, $k_2$, $\varepsilon_d$ and $d$ on the privacy amplification in FedPerm. Figure 5a shows the privacy amplification effect from $\varepsilon_d$-LDP to $(\varepsilon_\ell, \delta_\ell)$-DP for the local model updates after shuffling with $k_2$ shuffling patterns each with size of $k_1$. We can see that each client can use larger shuffling patterns (i.e. , increasing $k_1$) or more shuffling patterns (i.e., increasing $k_2$) and get larger privacy amplification. However, this comes with a price where this imposes more computation/communication burden on the clients to create the PIR queries as they need to encrypt $k_2 \times k_1^2$ values and send them to the server, and it also imposes higher computation on the server as it should multiply larger matrices. Figure 5b shows the amplification of privacy for fixed value of $k_1 = 100, k_2 = 10$ for various model sizes. From this figure we can see that if we want to provide same privacy level for larger models, we need to increase values of $k_1$ or $k_2$ (i.e. more computation/communication cost).

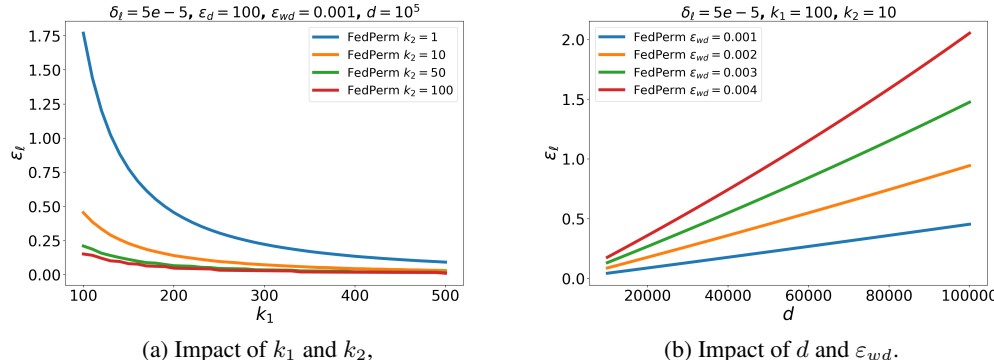

(a) Impact of $k_1$ and $k_2$,            (b) Impact of $d$ and $\varepsilon_{wd}$.

Figure 5: Privacy amplification of FedPerm from $\varepsilon_d$-LDP to $(\varepsilon_\ell, \delta_\ell)$-DP. We illustrate the overall amplification with Bennett inequality for the Laplace Mechanism.

## B   Missing Details of FedPerm Correctness

Note that for every client $u$ and every round $t$, decrypting the multiplication of the encrypted binary masks to the shuffled parameters produces the original unshuffled parameters. It means that for $y_u^t = \text{DEC}\left(c_u^t \times \tilde{y}_u^t\right)$. So for any $(\tilde{y}, c, )$ we have:

$$
\begin{aligned}
&\text{DEC}\left(c \times \tilde{y}\right) = \\
&\text{DEC}\left(\begin{bmatrix} \text{ENC}(\vec{b}_{\pi_1}) \\ \text{ENC}(\vec{b}_{\pi_2}) \\ \dots \\ \text{ENC}(\vec{b}_{\pi_d}) \end{bmatrix} \times \begin{bmatrix} \tilde{y}_1 \\ \tilde{y}_2 \\ \dots \\ \tilde{y}_d \end{bmatrix}\right) = \\
&\text{DEC}\left(\begin{bmatrix} \text{ENC}[0] & \dots & \text{ENC}[1] & \dots & \text{ENC}[0] \\ \text{ENC}[0] & \dots & \text{ENC}[1] & \dots & \text{ENC}[0] \\ \dots & \dots & \dots & \dots & \dots \\ \text{ENC}[0] & \dots & \text{ENC}[1] & \dots & \text{ENC}[0] \end{bmatrix} \times \begin{bmatrix} y_1^\pi \\ y_2^\pi \\ \dots \\ y_d^\pi \end{bmatrix}\right) = \\
&\text{DEC}\left(\begin{bmatrix} \text{ENC}[y_1] & \text{ENC}[y_2] & \dots & \text{ENC}[y_d] \end{bmatrix}\right) = \\
&\begin{bmatrix} y_1 & y_2 & \dots & y_d \end{bmatrix}
\end{aligned}
\tag{2}
$$

**Correctness of Aggregation:** In Equation 2, we show that $\forall t \in [T], u \in U \; y_u^t = \text{DEC}\left(c_u^t \times \tilde{y}_u^t\right)$. Based on the two main properties of a HE system **(a)** $m_1 \times m_2 \leftarrow \text{DEC}\left(\text{ENC}[m_1] \times m_2\right)$, **(b)** $m_1 + m_2 \leftarrow \text{DEC}\left(\text{ENC}[m_1] + \text{ENC}[m_2]\right)$, and Equation 2, we can derive the Equation 3:

$$\text{DEC}\left(\frac{1}{n}\sum_{u \in U}(c_u^t \times \tilde{y}_u^t)\right) = \frac{1}{n}\sum_{u \in U} y_u^t \tag{3}$$

## C   Missing Experiments

### C.1   Time Analysis

We evaluate the impacts of our hyperparameters $k_1$, $k_2$, $n$, and $d$s on the encryption, decryption and sever aggregation time in Figure 6. We use Paillier encryption system and we use a key size of $2048$ bits in our experiments. For measuring time, we use 64CPUs and 64GB memory for the client and server simulations. Note that we opt to not use GPU as model training is not a bottleneck in our system compared to HE operations. Also note that these figures are data independent as we are working with encryption and decryption and homomorphic multiplication with plaintext and homomorphic addition.

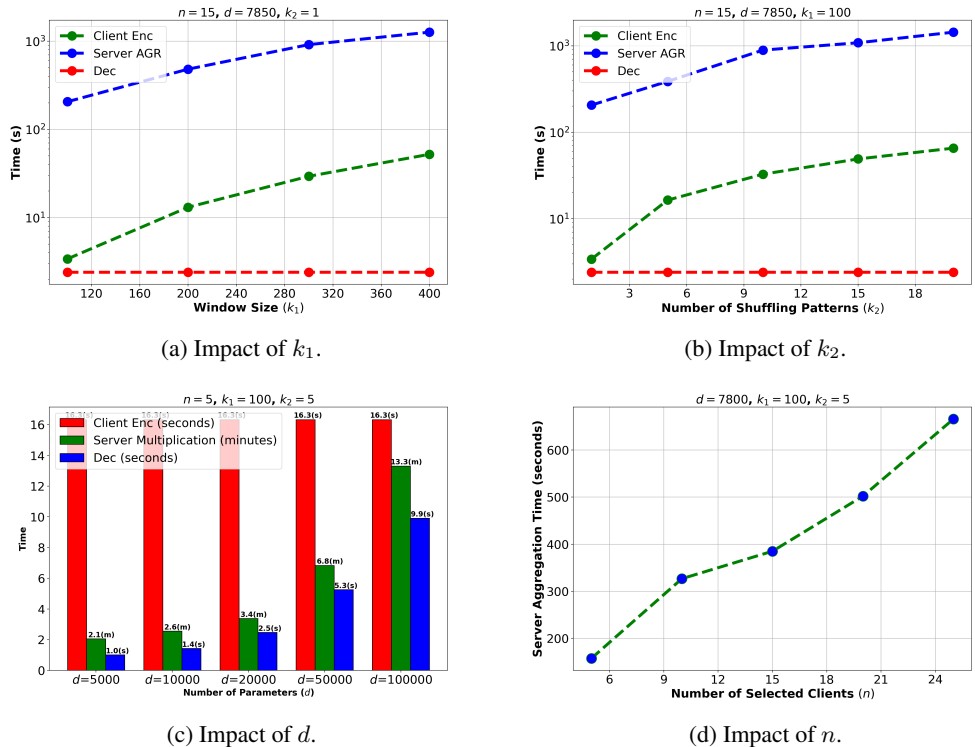

(a) Impact of $k_1$.

(b) Impact of $k_2$.

(c) Impact of $d$.

(d) Impact of $n$.

Figure 6: Client encryption, decryption, and server aggregation time in FedPerm.

**Client encryption time:**   In FedPerm, each client must do $k_1^2 \cdot k_2$ encryptions for its query, therefore client encryption time has a quadratic and linear relationship with window size ($k_1$) and number of shuffling patterns ($k_2$) respectively (Figures 6a and 6b). We also show in Figure 5 that increasing the $k_1$ has more impact (close to quadratic impact) compared to increasing $k_2$ on the privacy amplification. This means that if we invest more computation resources on the clients and are able to do more encryption, we get greater privacy amplification by parameter shuffling. For instance, when we increase the $k_1$ from 100 to 200 (while fixing $k_2 = 1$), the average client encryption time increases from 3.4 to 13.1 seconds for $d = 7850$ parameters. And while fixing the $k_1 = 100$, if we increase the number of shuffling patterns from 1 to 10, the encryption time goes from 3.4 to 32.7 seconds. When we fix the value of $k_1$ and $k_2$, the number of encryption is fixed at the clients, so the encryption time would be constant if we increase the number of parameters ($d$) each round (Figure 6c).

**Client decryption time:**   Changing $k_1$, $k_2$, and $n$ does not have any impact on decryption time, as each client should decrypt $d$ parameters (Figures 6a and 6b). In figure 6c, we show the linear

relationship of decryption time and number of parameters. For instance by increasing the number of parameters from $10^5$ to $10^6$, the decryption time increases from $1.01$ to $9.91$ seconds.

**Server aggregation time:** In FedPerm, the server first multiplies the encrypted binary mask to the corresponding shuffled model parameters for each client participating in the training round, and then sums the encrypted unshuffled parameters to compute the encrypted global model. We employ joblib to parallelize matrix multiplication over superwindows. Thus, larger the superwindows greater is the parallelism. However, as we increase $k_1$ and/or $k_2$ the superwindow size goes down, and hence the parallelism, which leads to increase in running time as observed in Figures 6a and 6b. Moreover, increasing $n, d$ has a linear relationship with server aggregation time (Figure 6c and 6d). For instance, when we increase the $n$ from 5 to 10 the server aggregation time increases from $157.47$ to $326.72$ seconds for $d = 7850$, $k_1 = 100$, and $k_2 = 1$.

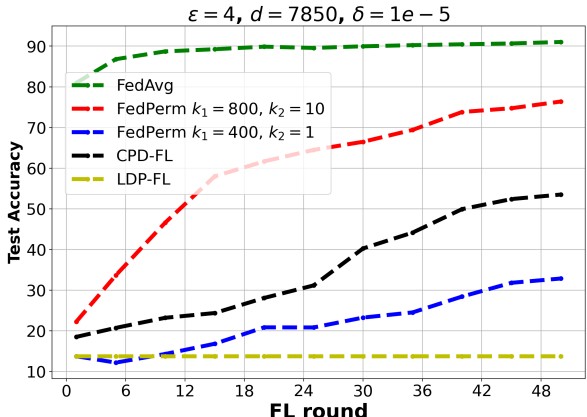

Figure 7: Test accuracy per FL round for different FL algorithms for MNIST over 15 clients.

## C.2   Accuracy per FL round

In Figure 7, we show the test accuracy for different algorithms per FL round when the total privacy budget is fixed to $\varepsilon = 4.0$.

# D   Threat Models

In this section, we describe two threat models that are of interest to our work, and illustrated in Figure 8.

## D.1   Honest-but-Curious Aggregator

In this threat model, we assume that the server correctly follows the aggregation algorithm, but may try to learn clients' private information by inspecting the model updates sent by the participants. This is a common assumption that previous works [53, 51, 11, 44] also consider. For creating the PIR queries, we use Paillier [39] homomorphic encryption. We explain different homomorphic encryption systems that we use in Appendix E.6. Before starting FedPerm, we need a key server to generate and distribute the keys for the homomorphic encryption (HE). A key server generates a pair of public and secret homomorphic keys $(Pk, Sk)$, sends them to the clients, and sends only the public key to the server. Either a trusted external key server or a leader client can be responsible for this role. For the leader client, similar to previous works [53], before the training starts, the FL server randomly selects a client as the leader. The leader client then generates the keys and distributes them to the clients and the server as above.

The steps of FedPerm for this threat model (Figure 8(a)) are as follows: **(1)** The pair of keys are distributed by the key server to all the clients. **(2)** In each round of training, the clients learn their local updates, generate encrypted PIR query and shuffled parameters, and send them to the server.

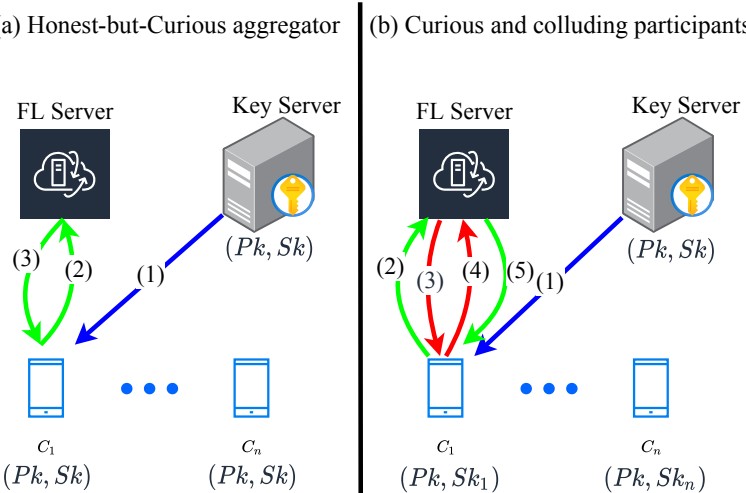

Figure 8: Different threat models.

Next, the server aggregates the updates, and sends the aggregated update to the clients. **(3)** Each client can decrypt the encrypted global parameters received from the server using the private key and updates its model.

### D.2 Curious and Colluding Clients

In this threat model, we assume that some clients may collude with the FL server to get private information about a victim client by inspecting its model update. For this threat model, we use thresholded Paillier [15]. In the thresholded Paillier scheme, the secret key is divided to multiple shares, and each share is given to a different client. For this threat model, we need an external key server that generates the keys and sends $(Pk, Sk_i)$ to each client, and sends the public key to the server. Now each client can partially decrypt an encrypted message, but if less than a threshold, say $t$, combine their partial decrypted values, they cannot get any information about the real message. On the other hand, if we combine $\geq t$ partial decrypted values, we can recover the secret. We explain how thresholeded Paillier scheme works in Appendix E.6.

The steps of FedPerm for this threat model (Figure 8(b)) are as follows: **(1)** The pairs of keys are distributed to the clients by the key server. **(2)** In each round of training, the clients learn their local updates, generate encrypted PIR query and shuffled parameters, and send them to the server. Next, the server aggregates the local updates to produce global model update (which is encrypted). **(3)** The server randomly chooses $t$ clients to partially decrypt the global model update. The FedPerm server sends the encrypted global update to these clients. **(4)** Each client decrypts the global model with its specific partial secret key $Sk_i$, and sends the result back to the server. **(5)** The server first authenticates each partial decryption that is done by the true $Sk_i$ (by a zero-knowledge proof provided by thresholded Paillier [15]). Then the FedPerm server combines the partial decrypted updates and broadcasts plain unshuffled model updates to all the clients for the next round of FedPerm.

At present our implementation of FedPerm does not support this threat model, and we leave it for future work.

## E  Background

### E.1  Central Differential Privacy in FL (CDP-FL)

In CDP-FL, the server provides participant-level DP by the perturbation. Formally, consider *adjacent* datasets ($X, X' \in \mathbb{R}^{n \times d}$) that differ from each other by the data of one federation client, then:

**Definition E.1 (Centralized Differential Privacy (CDP))** *A randomized mechanism $\mathcal{M} : \mathcal{X} \to \mathcal{Y}$ is said to be $(\varepsilon, \delta)$-differential private if for any two adjacent datasets $X, X' \in \mathcal{X}$, and any set*

$Y \subseteq \mathcal{Y}$:

$$\Pr[\mathcal{M}(X) \in Y] \le e^{\varepsilon} \Pr[\mathcal{M}(X') \in Y] + \delta \tag{4}$$

where $\varepsilon$ is the privacy budget (lower the $\varepsilon$, higher the privacy), and $\delta$ is the failure probability.

Algorithm 2 shows how CDP-FL works which is also discussed in [12, 22, 36]. In CDP-FL, the server receives model updates capped by norm $C$, and after averaging them, it adds i.i.d sampled noise to the parameters $\theta_g^{t+1} \leftarrow \theta_g^t + \frac{1}{n} \sum_{u \in U} \theta_u^t + \mathcal{N}(0, \sigma^2 \mathbb{I})$ where $\sigma \leftarrow \frac{\Delta_2}{\varepsilon} \sqrt{2ln(1.25)/\delta}$ and the $\ell_2$ sensitivity is $\Delta_2 = C$.

---

**Algorithm 2** Central Differential Privacy in FL (CDP-FL)

---

**Input**: number of FL rounds $T$, number of local epochs $E$, number of all the clients $N$, number of selected users in each round $n$, total privacy budget $TP$, probability of subsampling clients $q$, learning rate $\eta$, noise scale $z$, bound $C$
**Output**: global model $\theta_g^T$

1: $\theta_g^0 \leftarrow$ Initialize weights
2: Initialize MomentAccountant$(\varepsilon, \delta, N)$
3: **for** each iteration $t \in [T]$ **do**
4:    $U \leftarrow$ set of $n$ randomly selected clients out of $N$ total clients with probability of $q$
5:    $p_t \leftarrow$ MomentAccountant.getPrivacySpent() {% privacy budget spent till this round}
6:    **if** $p_t > TP$ **then**
7:      **return** $\theta_g^T$ {% if spent privacy budget is passed over the threshold finish FL training}
8:    **end if**
9:    **for** $u$ in $U$ **do**
10:      $\theta \leftarrow \theta_g^t$
11:      **for** local eopoch $e \in [E]$ **do**
12:        **for** batch $b \in [B]$ **do**
13:          $\theta \leftarrow \theta - \eta \nabla L(\theta, b)$
14:          $\triangle \leftarrow \theta - \theta_g^t$
15:          $\theta \leftarrow \theta_g^t + \triangle \min\left(1, \frac{C}{||\triangle||_2}\right)$
16:        **end for**
17:      **end for**
18:      Client $u$ sends $\theta_u^t = \theta - \theta_g^t$ to the server
19:    **end for**
20:    $\sigma \leftarrow zC/q$
21:    $\theta_g^{t+1} \leftarrow \theta_g^t + \frac{1}{n} \sum_{u \in U} \theta_u^t + \mathcal{N}(0, \sigma^2 \mathbb{I})$
22:    MomentAccountant.accumulateSpentBudget$(z)$
23: **end for**
24: **return** $\theta_g^T$

---

### E.2 Laplace Mechanism

The most common mechanism for achieving pure $\varepsilon_\ell$-DP is Laplace mechanism, where

**Definition E.2** *Let $f : \mathcal{X}^n \to \mathbb{R}^k$. The $\ell_1$-sensitivity of $f$ is:*

$$\Delta_1^{(f)} = \max_{X,X'} ||f(X) - f(X')||_1 \tag{5}$$

*where $X, X' \in \mathcal{X}^n$ are neighboring datasets differing from each other by a single data record.*

Sensitivity gives an upper bound on how much the output of the randomizer can change by switching over to a neighboring dataset as the input.

**Definition E.3** *Let $f : \mathcal{X}^n \to \mathbb{R}^k$. The Laplace mechanism is defined as:*

$$\mathcal{R}(X) = f(X) + [Y_1, \ldots, Y_k] \tag{6}$$

*Where the $Y_i$s are drawn i.i.d from Laplace$(\Delta^{(f)}/\varepsilon)$ random variable. This distribution has density of $p(x) = \frac{1}{2b} \exp\left(-\frac{|x|}{b}\right)$ where $b$ is the scale and equal to $\Delta^{(f)}/\varepsilon$.*

In FedPerm, each client $i$ applies the Laplace mechanism as a randomizer $\mathcal{R}$ on its local model update $(x_i)$. Each model update contains $d$ parameters in range of $[0, 1]$, so the sensitivity of the entire input vector is $d$. It means that applying $\varepsilon_d$-DP on the vector $x_i$ is equal to applying $\varepsilon_{wd} = \varepsilon_d/d$ on each parameter independently. Therefore, applying $\varepsilon_d$-DP randomizer $\mathcal{R}$ on the vector $x_i$ means adding noise from Laplace distribution with scale $b = \frac{1}{\varepsilon_{wd}} = \frac{1}{\frac{\varepsilon_d}{d}} = \frac{d}{\varepsilon_d}$.

## E.3 Privacy Composition

We use following naive and strong composition theorems [18] in this paper:

**Lemma E.1 (Näive Composition)** $\forall \varepsilon \geq 0, t \in \mathbb{N}$, the family of $\varepsilon$-DP mechanism satisfies $t\varepsilon$-DP under $t$-fold adaptive composition.

**Lemma E.2 (Strong Composition)** $\forall \varepsilon, \delta, \delta' > 0, t \in \mathbb{N}$, the family of $(\varepsilon, \delta)$-DP mechanism satisfies $(\sqrt{2t \ln(1/\delta')} \cdot \varepsilon + t \cdot \varepsilon(e^\varepsilon - 1), t\delta + \delta')$-DP under $t$-fold adaptive composition.

## E.4 Robustness to poisoning attacks

Most of the distributed learning algorithms, including FedAvg [32], operate on mutually untrusted clients and server. This makes distributed learning susceptible to the threat of poisoning [25, 41]. A *poisoning adversary* can either own or control a few of FL clients, called *malicious clients*, and instruct them to share malicious updates with the central server in order to reduce the performance of the global model. There are two approaches to poisoning FL: *untargeted* [7, 20, 40] attacks aim to reduce the utility of global model on arbitrary test inputs; and *backdoor* [3, 46, 50] attacks aim to reduce the utility on test inputs that contain a specific signal called the trigger.

In order to make FL robust to the presence of such malicious clients, the literature has designed various *robust aggregation rules (AGR)* [10, 33, 52, 35], which aim to remove or attenuate the updates that are more likely to be malicious according to some criterion. For instance, Multi-krum [10] repeatedly removes updates that are far from the geometric median of all the updates, and Trimmed-mean [52] removes the largest and smallest values of each update dimension and calculates the mean of the remaining values. It is not possible to use these AGRs in secure aggregation as the parameters are encrypted.

## E.5 Private Information Retrieval (PIR)

Private information retrieval (PIR) is a technique to provide query privacy to users when fetching sensitive records from untrusted database servers. That is, PIR enables users to query and retrieve specific records from untrusted database server(s) in a way that the servers cannot identify the records retrieved. There are two major types of PIR protocols. The first type is *computational PIR* (cPIR) [13] in which the security of the protocol relies on the computational difficulty of solving a mathematical problem in polynomial time by the servers, e.g., factorization of large numbers. Most of the cPIR protocols are designed to be run by a single database server, and therefore to minimize privacy leakage they perform their heavy computations on the whole database (even if a single entry has been queried). Most of these protocols use homomorphic encryption (Section E.6) to make their queries private. The second major class of PIR is *information-theoretic PIR* (ITPIR) [34]. ITPIR protocols provide information-theoretic security, however, existing designs need to be run on more than one database servers, and they need to assume that the servers do not collude. Our work uses computational PIR (cPIR) protocols to make the shuffling private.

## E.6 Homomorphic Encryption (HE)

Homomorphic encryption (HE) allows application of certain arithmetic operations (e.g., addition or multiplication) on the ciphertexts without decrypting them. Many recent works [13] advocate using partial HE, that only supports addition, to make the FL aggregation secure. In this section we describe two important HE systems that we use in our paper.

### E.6.1 Paillier

An additively homomorphic encryption system provides following property:

$$Enc(m_1) \circ Enc(m_2) = Enc(m_1 + m_2) \tag{7}$$

where $\circ$ is a defined function on top of the ciphertexts.

In these works, the clients encrypt their updates, send them to the server, then the server can calculate their sum (using the $\circ$ operation) and sends back the encrypted results to the clients. Now, the clients can decrypt the global model locally and update their models. Using HE in these scenario does not produce any accuracy loss because no noise will be added to the model parameters during the encryption and decryption process.

### E.6.2 Thresholded Paillier

In [15], the authors extend the Paillier system and proposed a thresholded version. In the thresholded Paillier scheme, the secret key is divided to multiple shares, and each share is given to a different participant. Now each participant can partially decrypt an encrypted message, but if less than a threshold, say $t$, combine their partial decrypted values, they cannot get any information about the real message. On the other hand, if we combine $\geq t$ partial decrypted values, we can recover the secret. In this system, the computations are in group $\mathbb{Z}_{n^2}$ where $n$ is an RSA modulus. The process is as follows:

- *Key generation:* First find two primes $p$ and $q$ that satisfies $p = 2p'+1$ and $q = 2q'+1$ where $p', q'$ are also prime. Now set the $n = pq$ and $m = p'q'$. Pick $d$ such that $d = 0 \mod m$ and $d = 1 \mod n^2$. Now the key server creates a polynomial $f(x) = \sum_{i=0}^{k-1} a_i x^i \mod n^2 m$ where $a_i$ are chosen randomly from $\mathbb{Z}_{n^2 m}^*$ and the secret is hidden at $a_0 = d$. Now each secret key share is calculated as $s_i = f(i)$ for $\ell$ shareholders and the public key is $n$. For verification of correctness of decryption another public value $v$ is also generated where the verification key for each shareholder is $v_i = v^{\Delta s_i} \mod n^2$ and $\Delta = \ell$.
- *Encryption:* For message $M$, a random number $r$ is chosen from $\mathbb{Z}_{n^2}^*$ and the output ciphertext is $c = g^M \cdot r^{n^2} \mod n^2$.
- *Share decryption:* The $i^{th}$ shareholder computes $c_i = c^{2\Delta s_i}$ for ciphertext $c$. And for zero-knowledge proof, it provides $\log_{c^4}(c_i^2) = \log_v(v_i)$ which provides guarantee that the shareholder really uses its secret share for decryption $s_i$.
- *Share combining:* After collecting $k$ partial decryption, the server can combine them into the original plain-text message $M$ by $c' = \Pi_{i \in [k]} c_i^{2\lambda_{0,i}^S} \mod n^2$ where $\lambda_{0,i}^S = \Delta \Pi_{i' \in [k], i' \neq i} \frac{-i}{i - i'}$. And use it to generate the $M$.

## F  Discussion and Future Work

**Utilizing recursion in cPIR.** A solution to reduce the number encryptions and upload bandwidth at the clients would be using recursion in our cPIR. In this technique, the dataset is represented by a $k_3$-dimensional hypercube, and this allows the PIR client to prepare and send $k_3 \sqrt[k_3]{d}$ encrypted values where $k_3$ would be another hyperparameter. For future work, we can use this technique and reduce the number of encryptions which makes the upload bandwidth consumption lower too. For instance, if we have one shuffling pattern $k_2 = 1$, the number of encryption decreases from $k_1 d$ to $k_3 \sqrt[k_3]{k_1 d}$.

**Plugging newer PIR protocol.** FedPerm utilizes cPIR for private aggregation, and in particular we use [13] which is based on Paillier. However, any other cPIR protocol can be used in FedPerm. For example, SealPIR [2] can be used to reduce the number of encryptions at the client. SealPIR is based on SEAL which is a lattice based fully HE. The authors show how to compress the PIR queries and achieve size reduction of up to $274\times$. We defer analyzing FedPerm with other cPIR schemes to future work.

**Combination of an external client shuffler for more privacy amplification.** For further privacy amplification, we can use an external shuffler that shuffles the $n$ sampled clients' updates similar to FLAME [30]. For future work, we can use double amplification by first shuffling the parameters at the clients (i.e., detaching the parameter values from their position) and then shuffle the client's updates at the external shuffler (i.e., detaching the updates from their client's ID).

