# OpenReview forum: "Private and Robust Federated Learning using Private Information Retrieval and Norm Bounding"
_NeurIPS.cc/2022/Workshop/Federated_Learning — FL-NeurIPS 2022 Poster_

### Official Review · Reviewer_tqq6 · 2022-10-17
**Interesting paper but could be better explained**

This paper proposes FedPerm, which uses norm bounding and intra-model parameter shuffling to ensure robustness of a federated learning model. FedPerm’s use of cryptographic aggregation of model updates can result in significant computational overhead, and the tradeoff between this computational overhead and accuracy is studied. Theoretical differential privacy results with the proposed intra-model parameter shuffling are derived. Experiments on MNIST show that FedPerm can achieve higher accuracy with the same privacy guarantees, compared to existing differential privacy-based federated learning methods.

+ The proposed defense method is new and interesting. It brings together several types of defenses that have been proposed in various attacks, which may be necessary in practice to defend against a variety of attacks.

+ The tradeoffs between computation, privacy, and accuracy are studied, and hyperparameters are embedded in the model that can control the achieved tradeoff between these performance metrics.

-- The model intuition could be better explained. For instance, why do intra-model shuffling and parameter normalization? This does augment the privacy significantly, but wouldn’t it also complicate aggregations, as the updates for different parameters would be combined by the central server? It’s thus not clear Corollary 3.1 can be directly derived from Corollary 2.1, and the explanations in the paper (understandably, given space constraints) are too high-level for me to tell if this aggregation effect has been accounted for.

-- Although FedPerm combines multiple defenses that have been proposed for both model poisoning and privacy inference attacks, the combination of these defenses seems fairly straightforward. It would be interesting to see a discussion of how these defenses might counteract and/or amplify each other’s effects, and how this is manifested in the experimental results.

-- It would be interesting to see the empirical computation vs. accuracy tradeoffs, instead of simply showing the accuracy per iteration. It is also not clear if the computation time results presented are over the entire training time or just over a single global round.

--All evaluations are conducted on the MNIST dataset, which is pretty simple but already requires hours to train. How generalizable is FedPerm to larger datasets with more complex models?

---

### Official Review · Reviewer_hcvB · 2022-10-17

The paper proposes to leverage intra-model parameter shuffling to improve the privacy guarantees of both Local Differential Privacy (LDP) and Central Differential Privacy (CDP).

Strengths:
- The method is clearly explained, in particular in Algorithm 1
- The time analysis focuses on an essential aspect of practical Federated Learning (round rudation and wall-time to convergence).

Weaknesses:
- The paper is rather hard to follow, with a lot of important parts deffered in the Appendix (like the correctness of the proposed FedPerm algorithm (Appendix B), or threat model (appendix D)). Moreover, the introduction is misleading since if refers to LDP solely, whereas the experimental results (Figure 2) also compare to CDP.
- Experimental setup: why do the authors use the Laplace mechanism instead of the (widespread) Gaussian mechanism (line 169)?

---

### Decision · Program_Chairs · 2022-10-20

Accept (Poster)